

# Relaxation and entropy generation
# after quenching quantum spin chains

**Máté Lencsés[1,2], Octavio Pomponio[3] and Gabor Takács[1,2]***

**1** BME Department of Theoretical Physics, H-1111 Budapest, Budafoki út 8., Hungary
**2** BME "Momentum" Statistical Field Theory Research Group,
H-1111 Budapest, Budafoki út 8., Hungary
**3** Dipartimento di Fisica e Astronomia dell'Università di Bologna, I-40127 Bologna, Italy

⋆ takacsg@eik.bme.hu

## Abstract

This work considers entropy generation and relaxation in quantum quenches in the Ising and 3-state Potts spin chains. In the absence of explicit symmetry breaking we find universal ratios involving Rényi entropy growth rates and magnetisation relaxation for small quenches. We also demonstrate that the magnetisation relaxation rate provides an observable signature for the "dynamical Gibbs effect" which is a recently discovered characteristic non-monotonous behaviour of entropy growth linked to changes in the quasi-particle spectrum.



# 1  Introduction

Quantum quenches were introduced as a paradigmatic protocol for studying non-equilibrium dynamics in isolated quantum many-body systems [1,2], and are now routinely engineered in experiments with trapped ultra-cold atoms [3–10]. In the simplest case the system is prepared in the ground state of a pre-quench Hamiltonian, which is then suddenly changed into a post-quench Hamiltonian at time $t = 0$, initiating a non-trivial time evolution. When the pre-quench and post-quench Hamiltonians are translationally invariant the quench is called global, and results in an initial state which is not an eigenstate under the post-quench Hamiltonian, and is a highly excited state with a spatially uniform finite energy density. Under appropriate conditions, the subsequent time evolution can be described by a quasi-particle picture [1], where the initial state acts as a source of quasi-particle pairs which are responsible for the propagation of correlations and entanglement throughout the system. The range of validity of this picture includes models with non-interacting quasi-particles, as well as some integrable models with fully elastic scattering. In more general (non-integrable) systems, the quasi-classical picture is expected to remain valid as long as the post-quench density is small, i.e. the mean separation between quasi-particles is not too small compared to the correlation length.

Entanglement can be quantified by several different measures. For the global quantum quench defined above the system is described by a time-evolving pure state $|\Psi(t)\rangle$. The von Neumann entanglement entropy of a subsystem $A$ is

$$S_A(t) = -\text{Tr}_A \rho_A(t) \log \rho_A(t), \tag{1.1}$$

where the reduced density matrix is defined as the partial trace over the complement $\bar{A}$ of the subsystem $A$:

$$\rho_A(t) = \text{Tr}_{\bar{A}} |\Psi(t)\rangle\langle\Psi(t)|. \tag{1.2}$$

In systems with short-range interactions, ballistic propagation of the quasi-particles together with the Lieb-Robinson velocity bound [11] leads to an initial linear growth of entanglement entropy of any large enough subsystem, which later saturates to a value proportional to the subsystem size [12]. The saturation value of the entanglement entropy is eventually identical to the usual thermodynamic entropy of the asymptotic stationary state [10, 12–14], and consequently the growth of entanglement entropy can be interpreted as a signal of the approach to equilibrium. For integrable models, an approach to compute the growth of entanglement entropy explicitly was recently developed in [14, 15].

A more general set of entanglement measures are the so-called Rényi entropies

$$S_A^{(n)}(t) = \frac{1}{1-n} \log \text{Tr} \rho_A^n(t), \tag{1.3}$$

which in the limit $n \to 1$ converge to the von Neumann entropy $S_A(t)$; the $n = 2$ case is also known as the purity of the reduced density matrix $\rho_A(t)$. At present there is no generalisation

of the approach of [14, 15] to compute the time evolution of Rényi entropies in integrable models and only the stationary values are known [16–18]. In contrast to the von Neumann entanglement entropy the Rényi entropies are not related directly to the thermodynamics of the system. Nevertheless, the quasi-particle picture suggests that the evolution of the Rényi entropies tracks closely that of the von-Neumann entanglement entropy, including the initial linear growth and later saturation. Contrary to the von Neumann entanglement entropy, the Rényi entropies can also be directly accessed by experiment [10, 19].

The approach to equilibrium can also be characterised by the rates of relaxation of physical observables, such as order parameters. It is natural to expect a close correspondence between the entropy growth rates $S_A^{(n)}(t)$ and the relaxation rates of expectation values of order parameters after the quench. Indeed, in the transverse field Ising spin chain it is possible to compute the Rényi entropy growth rates [20] and the magnetisation relaxation rate [21] explicitly, and for small post-quench density they are related by simple proportionality factors. The same relation between entropy growth and relaxation rates was found in the scaling Ising field theory in [22] when comparing Rényi entropy growth rates to the magnetisation relaxation rate computed in [23].

The relation between entropy growth and relaxation is interesting since magnetisation relaxation rates have a straightforward physical meaning and can be measured much more directly. Therefore it is interesting to study how the behaviour of entanglement entropy growth is reflected in Rényi entropy and especially relaxation rates. A recent example of such a relation is that dynamical confinement can limit entropy growth, which is accompanied by apparently undamped oscillations in magnetisation [24].

Another interesting effect is the "dynamical Gibbs effect" when the appearance of a new quasi-particle excitation leads to acceleration of entanglement entropy growth after a quantum quench, which was found in the quantum Ising and Potts spin chains [25, 26]. In this case, magnetisation relaxation rates provide an experimental signature of the effect instead of the entanglement entropy which is not directly observable.

Motivated by the above considerations, in this paper we set out to analyse the relation between entropy growth and relaxation in quantum quenches in the quantum Ising and Potts spin chains. We start in Section 2 by recalling exact results for quenches in the transverse field Ising spin chain and exhibit the exact ratios of Rényi entropy rates to the magnetisation relaxation rate for small post-quench density. Section 3 proceeds to the transverse field Potts spin chain which is non-integrable, therefore we approach it via numerical simulation. For small quenches the ratios of Rényi entropy rates to the magnetisation relaxation rate again turn out to be universal rational numbers, related to the symmetry properties of the order parameter and the replica symmetry underlying the computation of the Rényi entropy. Section 4 turns to quenches in the paramagnetic phase corresponding to switching on a longitudinal magnetic field, demonstrating that the relaxation rate provides a direct way for observation of the "dynamical Gibbs effect" . Finally we present our conclusions in Section 5. Some details are relegated to the appendix, with Appendix A describing our numerical simulation procedures, while Appendix B presents the results of further numerical simulations in figures.

## 2 Quenches in the transverse field Ising spin chain

The quantum Ising spin chain with transverse field is defined by the Hamiltonian

$$H(g) = -J \sum_{j=1}^{N} \left( \sigma_j^x \sigma_{j+1}^x + g \sigma_j^z \right), \tag{2.1}$$

where $\sigma_j^\alpha$ are the Pauli matrices at site $j$, and $J$ is taken to be positive. The model can be solved exactly by mapping the spin variables to Majorana fermion modes $\alpha_k$ with momenta $k = 2\pi r/N$ [1] [27, 28], in terms of which the Hamiltonian becomes

$$H(g) = \sum_k \epsilon_g(k)\alpha_k^\dagger \alpha_k + E_0(g),$$ (2.2)

with the dispersion relation

$$\epsilon_g(k) = 2J\sqrt{1 + g^2 - 2g\cos k}$$ (2.3)

and ground state energy

$$E_0(g) = -\frac{1}{2}\sum_k \epsilon_g(k).$$ (2.4)

We consider a quench protocol where the pre-quench Hamiltonian is given by a transverse Ising Hamiltonian with transverse field $g_0$, and the quench corresponds to switching the transverse field to a new value $g$ at $t = 0$. Then both the pre-quench and the post-quench models are described by free Majorana fermions $\tilde\alpha_k$ and $\alpha_k$, respectively, related by the Bogolyubov transformation

$$
\begin{aligned}
\tilde\alpha_k &= \cos\left(\frac{\Delta_k}{2}\right)\alpha_k + i\sin\left(\frac{\Delta_k}{2}\right)\alpha_{-k}^\dagger, \\
\cos\Delta_k &= \frac{g g_0 - (g + g_0)\cos k + 1}{\sqrt{1 + g^2 - 2g\cos k}\sqrt{1 + g_0^2 - 2g_0\cos k}}.
\end{aligned}
$$ (2.5)

The initial state can be written in terms of the post-quench eigenstates as

$$|\Psi_0\rangle \equiv |0; g_0\rangle = \frac{1}{\mathcal{N}}\exp\left\{i\sum_k K(k)\alpha_k^\dagger \alpha_{-k}^\dagger\right\}|0; g\rangle,$$ (2.6)

with

$$K(k) = \tan\left(\frac{\Delta_k}{2}\right) \quad, \quad \cos\Delta_k = \frac{1 - K^2(k)}{1 + K^2(k)},$$ (2.7)

which allows for a detailed evaluation of the time evolution [21].

## 2.1 Magnetisation

For a quench inside the ferromagnetic regime i.e. both $g, g_0 < 1$, the order parameter evolves in time according to [21]

$$m(t) \equiv \langle\Psi_0(t)|\sigma^x|\Psi_0(t)\rangle \simeq (\mathcal{C}_{FF})^{\frac{1}{2}}\exp\left\{\left[t\int_0^\pi \frac{dk}{\pi}\epsilon_g'(k)\ln|\cos\Delta_k|\right]\right\},$$ (2.8)

where $\mathcal{C}_{FF}$ was conjectured to be

$$\mathcal{C}_{FF} = \frac{1 - g g_0 + \sqrt{(1 - g^2)(1 - g_0^2)}}{2\sqrt{1 - g g_0}(1 - g_0^2)^{\frac{1}{4}}},$$ (2.9)

which follows by applying the cluster decomposition principle to the two-point function. The two-point function and the validity of (2.9) was recently reconsidered and a new result on $\mathcal{C}_{FF}$ is presented in [29]. Alternatively, a form factor expansion results in

$$m(t) \sim \sqrt{\xi}[1 + I(t) + O(t^{-1})]e^{-\Gamma t},$$ (2.10)

---

[1]The momentum quantum number $r$ can take either half-integer or integer values, corresponding to the Neveu-Schwarz and Ramond sectors.

where

$$\sqrt{\xi} = (1 - g^2)^{1/8} \equiv \langle 0; g | \sigma^x | 0; g \rangle$$

$$I(t) = A_0 \frac{\cos\left(2\epsilon_g(0)t + \frac{3\pi}{4}\right)}{t^{3/2}} - A_\pi \frac{\cos\left(2\epsilon_g(\pi)t - \frac{3\pi}{4}\right)}{t^{3/2}} + o(t^{-3/2}) \tag{2.11}$$

$$\Gamma = \int_0^\pi \frac{dk}{\pi} K^2(k)[2\epsilon_g'(k)] + O(K^6),$$

with

$$A_k = \frac{hJ^2 K'(k)}{\sqrt{\pi}\epsilon_g(k)^2 \left|\epsilon_g''(k)\right|^{3/2}}, \qquad k = 0, \pi, \tag{2.12}$$

which is equivalent to 2.8 in the limit where $K^2(k) \ll 1$.

In the scaling limit the relaxation rate $\Gamma$ in (2.11) is given by [23]

$$\Gamma = 2M \int_0^\infty \frac{d\theta}{\pi} \hat{K}^2(\theta) \sinh\theta, \tag{2.13}$$

where $M = 2J|1-h|$ is the fermion mass which is kept fixed in the scaling limit, and

$$i\hat{K}(\theta) = K(\theta) = i \tan\left[\frac{1}{2}\arctan(\sinh\theta) - \frac{1}{2}\arctan\left(\frac{M}{M_0}\sinh\theta\right)\right], \tag{2.14}$$

with $M_0 = 2J|1-h_0|$ denoting the fermion mass of the pre-quench system.

## 2.2 Rényi entropies vs. relaxation rate

### 2.2.1 In the scaling limit

Consider a spatial partition of the system into a subsystem $A$ its complement $\bar{A}$, the Rényi entropies are given by

$$S_n := \frac{1}{1-n}\log\operatorname{Tr}\rho_A^n. \tag{2.15}$$

Its evaluation can be handled by the replica trick where $n$ is the number of replicas and Rényi entropy for an interval can be represented as a correlator of branch-point twist fields [30, 31].

Quenching the transverse field corresponds to changing the mass parameter from a value $m_0$ to $m$ in the scaling Ising field theory (free Majorana fermion). For the case when the subsystem $A$ spans the (semi-infinite) left/right half of space, the Rényi entropies can be computed as the expectation values of the appropriate branch-point twist fields, which can be evaluated to $O(K^2)$ using form factors resulting in [22]

$$S_n(t) = S_n(0) + \frac{\Gamma n t}{2(n-1)} + \frac{n\mu^2}{64\pi M t(n-1)} + \frac{\mu}{8\sqrt{\pi}n}\frac{\cos\frac{\pi}{2n}}{\sin^2\frac{\pi}{2n}}\frac{\cos\left(2Mt - \frac{\pi}{4}\right)}{(n-1)(Mt)^{3/2}} + O(t^{-3}), \tag{2.16}$$

where

$$\mu = 1 - \frac{M}{M_0} \tag{2.17}$$

parameterises the quench magnitude. We remark that it is unclear at present how to carry out the $n \to 1$ limit to obtain the von Neumann entropy in the scaling field theory.

The result (2.16) implies that in the leading order in $\mu$ the following relation holds between the relaxation rate of the magnetisation and the growth rates of the $n > 1$ Rényi entropies:

$$\gamma_n^{TFIM} = \frac{\Gamma_n^{TFIM}}{\Gamma} = \frac{1}{2}\frac{1}{1-1/n}. \tag{2.18}$$

### 2.2.2 On the spin chain

In the case of a finite region $A$ of $L$ number of sites, one can calculate the eigenvalues of the reduced density matrix from a block Toeplitz matrix [12,32], which allow the computation of the $S_n$. For long times $t$, assuming that $L \gg t$ the time evolution of the entropies is given by [20]:

$$S_n(t) \approx 2S_n(0) + 2\Gamma_n^{TFIM} t, \qquad (2.19)$$

where $2S_n(0)$ is the entropy of the initial state and the growth rates $\Gamma_n^{TFIM}$ are given as:

$$\Gamma_n^{TFIM} = \frac{1}{1-n} \int_0^\pi \frac{dk}{\pi} |\epsilon_g'(k)| \log(P_n(\cos \Delta_k)), \qquad (2.20)$$

where $P_n(x) = (\frac{1+x}{2})^n + (\frac{1-x}{2})^n$. The factor of two in (2.19) is related to the number of boundaries between the subsystems $A$ and $B$, or identically the number of brach-point twist field insertions. This result holds for any transverse quenches regardless of the phase, but in order to compare it to the relaxation of magnetisation we restrict our consideration to the quenches within the ferromagnetic regime in the following.

In Table 2.1 we compute the ratios $\gamma_n^{TFIM}$ for quenches on the spin chain by numerical integration of (2.8) and (2.20) illustrating the relation (2.18). It is clear that (2.18) only holds approximately, and the agreement is better for smaller quenches. Note that a finite quench in the scaling field theory corresponds to a limit in the spin chain when the quench magnitude goes to zero, so the universal ratio (2.18) continues to hold to the lowest order of the quench amplitude $K$, i.e. for small post-quench density.

It is possible to show explicitly that (2.18) holds for small quasi-particle density for the discrete chain itself. At $O(K^2)$ we have that $P_n(\cos \Delta_k)$ as a function of $K$ is given by

$$P_n(\cos \Delta_k) = 1 - nK^2 + O(K^4) \qquad (2.21)$$

so that (2.20) becomes

$$\Gamma_n^{TFIM} = \frac{n}{n-1} \int_0^\pi \frac{dk}{\pi} |\epsilon_g'(k)| K^2(k) + O(K^4). \qquad (2.22)$$

Comparing the latter result with $\Gamma$ in (2.11) it is clear that the relation (2.18) holds at the leading order in $K$. Including the first correction leads to

$$\gamma_n^{TFIM} = \frac{1}{2} \frac{1}{1-1/n} \left( 1 - \frac{3n+1}{2} \frac{\int_0^\pi \frac{dk}{\pi} |\epsilon_g'(k)| K^4(k)}{\Gamma} \right). \qquad (2.23)$$

For the von Neumann entropy we have

$$\lim_{n \to 1} \frac{1}{1-n} \log P_n(x) = -\frac{1}{P_1(x)} \frac{\partial P_n(x)}{\partial n} \Big|_{n=1}$$
$$= \frac{1+x}{2} \log\left(\frac{1+x}{2}\right) + \frac{1-x}{2} \log\left(\frac{1-x}{2}\right) \qquad (2.24)$$

resulting in

$$\gamma_1^{TFIM} = \frac{1}{2} \left( 1 - \frac{\int_0^\pi \frac{dk}{\pi} |\epsilon_g'(k)| K^2(k) \log K^2(k) + O(K^4)}{\Gamma} \right), \qquad (2.25)$$

which is $K$ dependent. In agreement with the numerical data, this ratio increases for smaller quenches.

Table 2.1: Values of $\gamma_n^{TFIM}$ for a set of transverse quenches in the ferromagnetic phase of the TFIM, in units obtained by setting $J = 1$.

| $g_0 \to g$ | $\gamma_1^{TFIM}$ | $\gamma_2^{TFIM}$ | $\gamma_3^{TFIM}$ | $\gamma_4^{TFIM}$ |
|---|---|---|---|---|
| $0.3 \to 0.5$ | 2.85355 | 1 | 0.746474 | 0.66351 |
| $0.5 \to 0.3$ | 2.85355 | 1 | 0.746474 | 0.66351 |
| $1/3 \to 2/3$ | 2.18374 | 1 | 0.737744 | 0.6555 |
| $2/3 \to 1/3$ | 2.18374 | 1 | 0.737744 | 0.6555 |
| $0.5 \to 0.7$ | 2.56297 | 1 | 0.743885 | 0.661164 |
| $0.7 \to 0.5$ | 2.56297 | 1 | 0.743885 | 0.661164 |
| $0.6 \to 0.66$ | 3.74455 | 1 | 0.749385 | 0.666119 |
| $0.6 \to 0.66$ | 3.74455 | 1 | 0.749385 | 0.666119 |
| $0.6 \to 0.606$ | 6.10712 | 1 | 0.749996 | 0.666663 |
| $0.1 \to 0.11$ | 6.03714 | 1 | 0.749994 | 0.666661 |
| $0.1 \to 0.101$ | 8.34075 | 1 | 0.75 | 0.666666 |
| $0.3 \to 0.4$ | 3.60792 | 1 | 0.749189 | 0.665945 |
| $0.3 \to 0.2$ | 3.67499 | 1 | 0.74929 | 0.666035 |

# 3 Transverse quenches on the quantum Potts spin chain

## 3.1 The quantum Potts spin chain

The $q$-state quantum Potts spin chain consists of a chain of generalised spins having internal quantum states $|\mu\rangle_i$, with $i$ labeling the lattice sites and $\mu = 0, \dots, q-1$ the possible internal states of the spins, governed by the Hamiltonian

$$H = -J \left( \sum_i \sum_{\mu=0}^{q-1} P_i^\mu P_{i+1}^\mu + g \sum_i P_i \right). \tag{3.1}$$

The first term of the Hamiltonian contains the traceless projector $P_i^\mu = |\mu\rangle_i \langle\mu|_i$ which tends to align the spin at site $i$ along the direction $\mu$, while the second term is given by the traceless operator $P = |\lambda_0\rangle \langle\lambda_0| - 1/q$, which forces the spin along the direction $|\lambda_0\rangle \equiv \sum_\mu \frac{|\mu\rangle}{\sqrt{q}}$. The relative strength of these two terms is regulated by the transverse magnetic field $g$: $g > 1$ is the paramagnetic phase with a unique ground state, while $g < 1$ is the ferromagnetic phase with $q$ degenerate ground states, spontaneously breaking the global $\mathbb{S}_q$ symmetry. The case $q = 2$ is the quantum Ising spin chain, while $q = 3$ gives the 3-state quantum Potts spin chain which we call quantum Potts spin chain for short. In both of these cases, the partition function is invariant under the Kramers-Vannier duality $g \to 1/g$ [33], and the two phases are separated by a quantum phase transition at the critical point $g_c = 1$. For the quantum Ising spin chain, the spectral gap is given by $\Delta = 2J|1-g|$ (exact), while for the quantum Potts spin chain $\Delta \sim J|g-1|^{5/6}$ (for $g \sim 1$) where the exponent can be extracted from conformal field theory [34].

The quantum Potts spin chain is not integrable apart from the critical point. Nevertheless, its spectrum can easily be guessed, and subsequently verified using perturbation theory for $g$ far away from the critical value [35]. In the ferromagnetic phase, the elementary excitations are domain walls with dispersion relation

$$\epsilon^{\mu,\mu'}(k) = \epsilon_g(k), \tag{3.2}$$

where $\mu$ and $\mu'$ denote the orientations of domains linked by the excitation, with $\mu - \mu' = \pm 1$ mod

3, with the dispersion relation

$$\epsilon_g(k) = J\left(1 - \frac{2g}{3}\cos k\right) + O\left(g^2\right). \tag{3.3}$$

Conversely, for $g > g_c$ the ground state is non-degenerate and the elementary excitations are two kinds of local spin flips $\lambda = \pm$ with dispersion relation

$$\epsilon^\lambda(k) = \tilde{\epsilon}_g(k) \quad , \quad \tilde{\epsilon}_g(k) = 3J\left(1 - \frac{2}{3g}\cos k\right) + O\left(g^{-2}\right). \tag{3.4}$$

The two excitations are related to each other by the charge conjugation mapping the spin direction $\mu$ to $-\mu$ mod 3.

## 3.2 Quenches in the transverse field

We consider quantum quenches starting at time $t = 0$ from the ground state $|\Psi_0\rangle$ of a pre-quench Hamiltonian $H_0$ corresponding to a pre-quench value of transverse field $g_0$, evolved for $t > 0$ by the post-quench Hamiltonian $H$ corresponding to transverse field $g$:

$$|\Psi_0(t)\rangle = e^{-iHt}|\Psi_0\rangle. \tag{3.5}$$

For the quantum Potts spin chain there are no analytic results for time evolution since the model is not integrable. However, we expect a quench dynamics qualitatively similar to the Ising case, and so considering quenches inside the ferromagnetic phase $g, g_0 < 1$ we can assume that the magnetisation evolves according to

$$m_0(t) \equiv \langle\Psi_0(t)|P^0|\Psi_0(t)\rangle \simeq (\mathcal{C}_{FF}^{Potts})\exp\{-t\Gamma^{Potts}\}, \tag{3.6}$$

where $\mathcal{C}_{FF}^{Potts}$ and $\Gamma^{Potts}$ are some positive constants that can be determined fitting the iTEBD data. For Rényi entropies (computed for semi-infinite subsystem) we can again assume that time evolution occurs according to

$$S_n(t) = S_n(0) + \Gamma_n^{Potts}t, \tag{3.7}$$

where again $\Gamma_n^{Potts}$ can be obtained fitting the iTEBD data. Examples are shown in Fig. 3.1, while Table 3.1 summarizes the results for the ratios $\gamma_n^{Potts} = \Gamma_n^{Potts}/\Gamma^{Potts}$ for different values of $g_0$ and $g$.

Table 3.1: Values of $\gamma_n^{Potts}$ for various transverse quenches.

| $g_0 \to g$ | $\gamma_1^{Potts}$ | $\gamma_2^{Potts}$ | $\gamma_3^{Potts}$ | $\gamma_4^{Potts}$ |
|---|---|---|---|---|
| $0.3 \to 0.5$ | 4.20 | 1.34 | 1.01 | 0.89 |
| $0.5 \to 0.3$ | 4.21 | 1.34 | 1.01 | 0.89 |
| $1/3 \to 2/3$ | 3.29 | 1.32 | 1.00 | 0.89 |
| $2/3 \to 1/3$ | 3.34 | 1.32 | 1.00 | 0.89 |
| $0.5 \to 0.7$ | 3.82 | 1.32 | 1.00 | 0.88 |
| $0.7 \to 0.5$ | 3.88 | 1.33 | 1.00 | 0.89 |

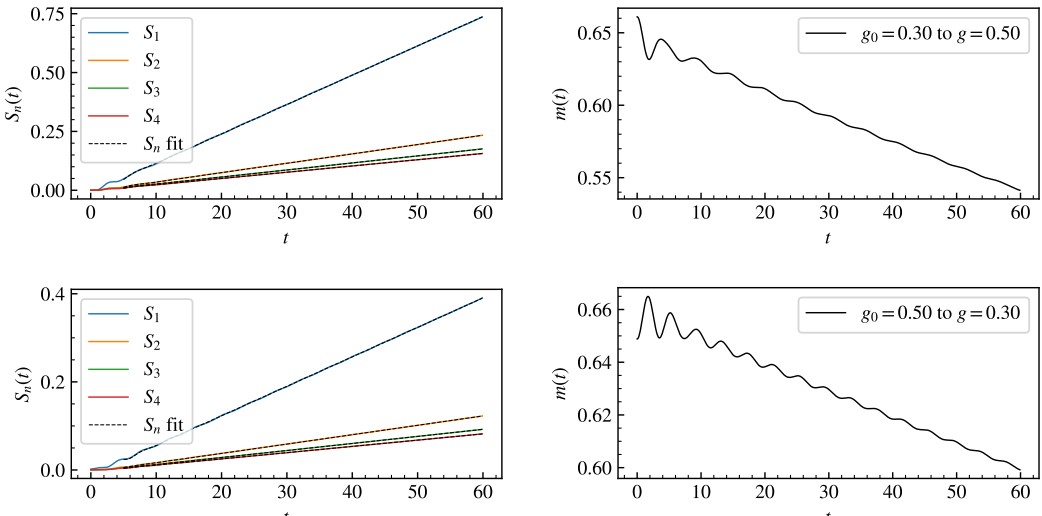

Figure 3.1: Rényi entropies and magnetisation time evolution for the quenches $g_0 = 0.3 \to g = 0.5$ (a) and $g_0 = 0.5 \to g = 0.3$ (b). Time is measured in units of $1/J$, and the initial entropy was subtracted.

## 3.3 Digression: global symmetries, twist fields and replicas

In general, quantum field theories can have global internal symmetry transformations $\{\phi\} \to \{\mathcal{R}\phi\}$ acting on the fields of the field theory. Twist fields, or symmetry fields are defined in the path integral formalism: a twist field insertion $\langle \mathcal{T}_{\mathcal{R}} \ldots \rangle$ changes the boundary condition for the path integral along the cut starting from the insertion point to:

$$\{\phi^+\} = \{\mathcal{R}\phi^-\}, \tag{3.8}$$

where $\{\phi^{\pm}\}$ denote the set of fields on the two sides of the cut. From this definition it is clear that taking a local operator $\mathcal{O}$ around the twist-field insertion results in the change $\mathcal{O} \to \mathcal{R}\mathcal{O}$. Therefore the correlator $\langle \mathcal{T}_{\mathcal{R}} \mathcal{O} \ldots \rangle$ is not a single valued function if $\mathcal{R}\mathcal{O} \neq \mathcal{O}$, and $\mathcal{O}$ is said to be "semi-local" with respect to $\mathcal{T}_{\mathcal{R}}$.

The $q$-state Potts model has the permutation symmetry $\mathbb{S}_q$ of the spins, with the magnetisation field as order parameter. In the scaling field theory the kink excitations are created by the disorder operators [36] which are semi-local with respect to the magnetisation [37] and the respective symmetry is given by the cyclic subgroup $\mathbb{Z}_q$. Note that the same properties hold for the lattice model [38].

The Rényi entropies can be constructed using the so-called replica trick [12,30,39]. Considering the scaling field theory limit, the $n$-th power of the reduced density matrix of a finite interval can be represented as a path integral over an $n$-sheeted Riemann surface with each sheet corresponding to a replica of the original QFT. The branching points of the Riemann surface can be represented as the insertion points of branch-point twist-fields. The Rényi entropy of an interval is proportional to the logarithm of a branch-point twist-field two-point function, while the bipartite Rényi entropy of a system cut in two semi-infinite halves can be obtained from a one-point function.

The replica theory has a $\mathbb{S}_n$ replica symmetry in addition to the possible global symmetries acting within the replicas, and the original QFT fields describing observables and creating the particle excitations have a copy inside each replica. These copies are transferred to the next Riemann sheet when taken around a branch-point twist-field, which corresponds to the action by the generator of the cyclic group $\mathbb{Z}_n$ [30]. Therefore the branch-point twist-field acts as a $\mathbb{Z}_n$

twist field for the fields creating the particle excitations, which entirely parallels the situation for the magnetisation and its associated $\mathbb{Z}_q$ symmetry.

### 3.4 Conjecture for the universal ratio

From the numerical results one can form the following conjecture: for $n \neq 1$

$$\gamma_n^{Potts} = \frac{2}{3} \frac{1}{1-1/n}, \tag{3.9}$$

in the limit of small quenches. Together with (2.18) this result hints that for a generic $q$-state quantum Potts spin chain

$$\gamma_n^{(q)} = \frac{1-1/q}{1-1/n}. \tag{3.10}$$

This conjecture can be heuristically supported as follows. Note that the Rényi entropy is associated to a replica trick, which has a symmetry $\mathbb{Z}_n$. The replica structure can be implemented by a branch-point twist field, which is semi-local with respect to the intertwining fields creating the particles in the different replicas [30] and the entropy growth rate itself is nothing else than the relaxation rate of the expectation value of the $\mathbb{Z}_n$ twist field divided by $n-1$.

Turning to the magnetisation, it is a field with $\mathbb{Z}_q$ symmetry. The role of replicas is played by the sectors built upon the $q$ different ground states. In this case the roles are reversed: the magnetisation is a local field acting within a given sector, while the particle excitations are kinks (domain walls) created by disorder operators [40, 41], however their mutual semi-locality properties are analogous to the case of the branch-point twist fields. In fact, the factor $\frac{1}{1-1/q}$ appears in the annihilation pole of the two-kink form factor of the magnetisation [41]. It is exactly the kinematical poles that give rise to the secular terms in the time evolutions which can be resummed to yield the relaxation of the magnetisation [23], which explains the $q$-dependence in (3.10).

It is then tempting to argue that the full ratio (3.10) is just a result of the symmetry properties, whereby the relaxation rate for the branch-point twist field can be obtained from that of the magnetisation by replacing $q$ with $n$. There is a caveat, however: the Rényi entropy growth rate depends on its normalisation, i.e. the choice of the prefactor in its definition (2.15). The prefactor is necessary to recover the von Neumann entropy in the limit $n \to 1$, however, it is not so obvious how to argue for $n > 1$. Therefore, the argument can only be made robust by comparing the computation of the relaxation rates of the branch-point twist fields to that of the magnetisation in the $q$-state Potts model in more detail, and we return to this issue in the Conclusions.

Similarly to the Ising case, the universal ratio (3.10) is only expected to be valid when considering small quenches. To demonstrate this numerically for the 3-state Potts chain, we considered quenches starting from $g_0 = 0.7$ to different values of $g$ (see Fig. 3.2). These results are consistent with the analytic considerations presented for the Ising chain in subsection 2.2: the ratios $\gamma_n^{Potts}$ start to deviate from (3.10) once $|g - g_0|$ (and therefore the post-quench quasi-particle density) is increased.

## 4 Longitudinal quenches in the paramagnetic phase

The other class of quenches we consider corresponds to starting from a transverse spin chain and switching on a *longitudinal* field $h$. For the quantum Ising spin chain this means a quench from

$$H(g,0) = -J \sum_{j=1}^{N} \left( \sigma_j^x \sigma_{j+1}^x + g \sigma_j^z \right), \tag{4.1}$$

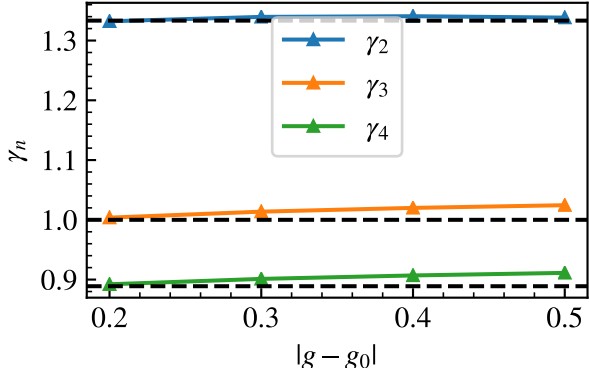

Figure 3.2: Values of $\gamma_n$ quenching from $g_0 = 0.7$ to $g = 0.5, 0.4, 0.3, 0.2$. The dashed lines represent the conjectured ratios (3.10).

to

$$H(g,h) = -J \sum_{j=1}^{N} \left( \sigma_j^x \sigma_{j+1}^x + g\sigma_j^z + h\sigma_j^x \right). \tag{4.2}$$

In the ferromagnetic phase this leads to confinement [24] which limits entropy growth and prevents equilibration by truncating the spread of correlations; therefore here we only consider these quenches in the paramagnetic phase.

These quenches lead to the dynamical manifestation of the Gibbs mixing entropy observed in [25, 26], resulting in a non-monotonic behaviour of the von Neumann entropy growth rate with the quench magnitude $h$. We demonstrate that the same behaviour is reflected in the relaxation rate of magnetisation and the Rényi entropy growth rates, which opens the way for an experimental observation of the "dynamical Gibbs effect".

## 4.1 Ising

We present the numerical results for longitudinal quenches in the quantum Ising spin chain with transverse field $g = 1.75$, leaving the transverse field unchanged and quenching the longitudinal field from 0 to $h$ in Fig. 4.1. Some plots for different quenches showing the same behaviour are relegated to Appendix B.

In this case we do not expect that the ratio

$$\gamma_n = \frac{\Gamma_n}{\Gamma} = \frac{1}{2} \frac{1}{1 - 1/n} \tag{4.3}$$

holds in the limit of small quenches, since the $\mathbb{Z}_2$ symmetry characterising the magnetisation is broken by the longitudinal field $h$. However, the replica symmetry of the twist fields associated to the Rényi entropies is unaffected, and therefore we expect that

$$\frac{\Gamma_n}{\Gamma_m} = \frac{1 - 1/m}{1 - 1/n}, \tag{4.4}$$

which is in fact the case as shown in Fig. 4.1.

Note that the characteristic minimum and subsequent fast growths displayed by the von Neumann entropy growth rate $\Gamma_1$ is followed very well by the Rényi entropy rates $\Gamma_n$, with the maximum and the subsequent minimum at the same location within a very good approximation. We recall that the critical value of the longitudinal field $h \approx 0.4$ where the rates take

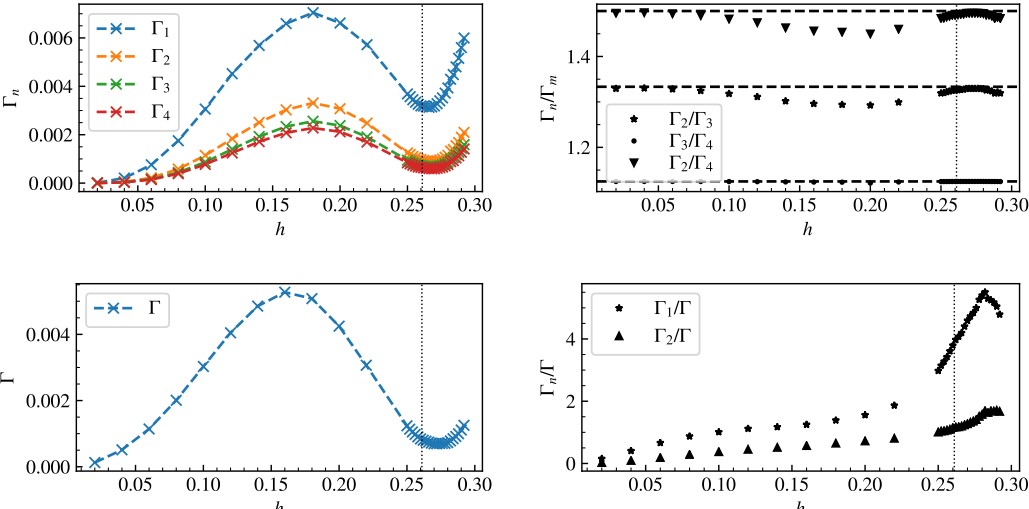

Figure 4.1: Left panels: Entropy growth rates $\Gamma_n$ (top) and magnetisation relaxation rate $\Gamma$ (bottom) in units $J = 1$ for longitudinal quenches in the quantum Ising spin chain starting from $g = 1.75$ as a function of the longitudinal coupling. Right: Ratios $\Gamma_n/\Gamma_m$ (top) and $\Gamma_n/\Gamma$ (bottom). The dashed lines represent the universal ratios (4.4) that are restored in the limit $h \to 0$. The vertical line is drawn at the value $h_{\text{crit}}$ where the second quasi-particle appears in the post-quench spectrum.

their minimum, coincides with the threshold for the appearance of a bound state quasi-particle in the spectrum, and the subsequent much faster generation of entropy can be associated to the contribution of Gibbs mixing entropy [25]. Additionally, the magnetisation relaxation rate follows the behaviour of the entropy growth rate, which means that it can be used as an experimental signal for the "dynamical Gibbs effect".

More precisely, the threshold value $h_{\text{crit}}$ where the new quasi-particle appears generally somewhat differs from the position $h_{\text{min}}$ of the minimum in the von Neumann entropy rate $\Gamma_1$. As already explained in [25] this is mainly due to two effects. Firstly, the bound state threshold value $h_{\text{crit}}$ is determined from the excitation spectrum above the ground state, while the post-quench system has a finite energy density which is expected to induce shifts in the effective quasi-particle masses. Secondly, the contribution which makes $\Gamma_1$ grow for $h > h_{\text{crit}}$ must grow sufficiently in size to counteract the one that made it decrease for $h < h_{\text{crit}}$. The latter effect is expected to depend on the quantity considered, i.e. one expects to find slightly different positions of the minima for $\Gamma_n$ for different $n$, as well as for the magnetisation relaxation rate $\Gamma$. However these differences are quite small and the positions of the local minimum (and similarly that of the local maximum) only depend very mildly on the rate considered.

Another interesting observation is that the quench corresponding to the critical longitudinal field field $h = h_{\text{crit}}$ also seems to be small in the sense that the ratios $\Gamma_n/\Gamma_m$ return close to the universal values (4.4) characteristic for small quenches. This indicates that somehow these quenches are also small as indicated by the slow growth of entanglement entropy, and that the proper condition for the universal ratios (4.4) to hold is slow growth of entropy rather than small post-quench density.

Finally we note that simulations for other values of the transverse field lead to the same conclusions, as shown in Figs. B.1, B.2 and B.3 for $g = 1.25$, $g = 1.5$ and $g = 2.00$, respectively.

## 4.2 Longitudinal quenches in the Potts spin chain

In this subsection we present the numerical results for longitudinal quenches in the quantum Potts spin chain. As shown in [26], this model shows effects of changes of the quasi-particle spectrum on the entropy growth rate as observed in the quantum Ising spin chain [25]. Just as for the Ising case, we consider four different values of transverse field ($g = 1.25, 1.50, 1.75, 2.00$) and performed a quench by adding a longitudinal field leaving the transverse field unchanged, i.e. starting from

$$H(g, 0) = -J \left( \sum_i \sum_{\mu=0}^{q-1} P_i^\mu P_{i+1}^\mu + g \sum_i P_i \right) \tag{4.5}$$

to

$$H(g, h) = -J \left( \sum_i \sum_{\mu=0}^{q-1} P_i^\mu P_{i+1}^\mu + g \sum_i P_i + \sum_i h P_i^{\mu_0} \right), \tag{4.6}$$

where the direction $\mu_0$ of the longitudinal field $h$ can eventually be chosen arbitrarily among the three possibilities 0, 1 and 2 without altering the physical behaviour.

For $g = 1.75$, the results are shown in 4.2, while the other three cases are presented in Figs. B.4, B.5 and B.6. Apart from difference in quantitative details such as the values of the critical longitudinal field and the numerical values of the rates themselves, the qualitative features and the essential conclusions are exactly the same as for the quantum Ising spin chain: (1) the universal ratios (4.4) hold both for small values of $h$ and, to a very good approximation for quenches around the critical value corresponding to the appearance of a new quasi-particle; and (2) the magnetisation relaxation rate closely follows the behaviour of the von Neumann and Rényi entropy rates, which can again be used as an observable signature for the "dynamical Gibbs effect".

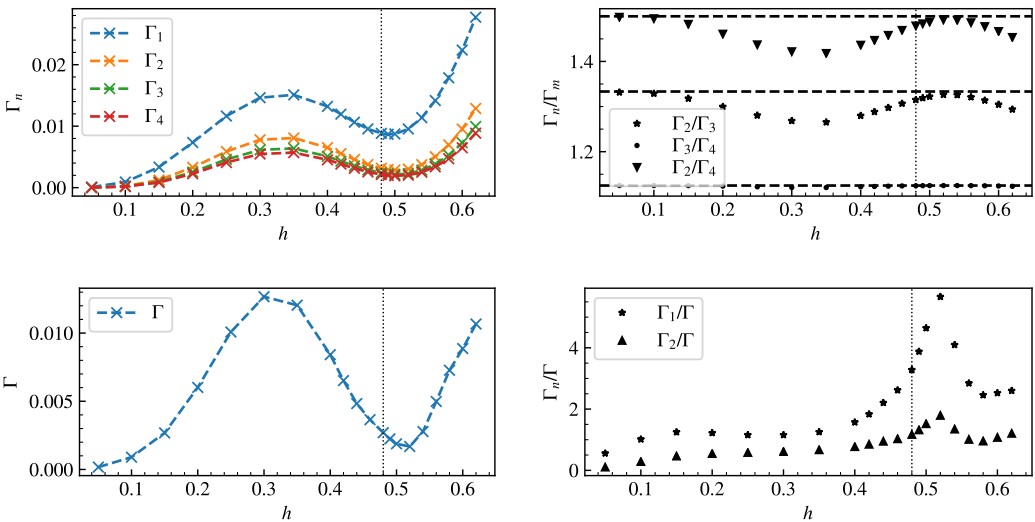

Figure 4.2: Left panels: Entropy growth rates $\Gamma_n$ (top) and magnetisation relaxation rate $\Gamma$ (bottom) for longitudinal quenches in the quantum Potts spin chain starting from $g = 1.75$ as a function of the longitudinal coupling, in units with $J = 1$. Right: Ratios $\Gamma_n/\Gamma_m$ (top) and $\Gamma_n/\Gamma$ (bottom). The dashed lines represent the universal ratios (4.4) that are restored in the limit $h \to 0$. The vertical line is drawn at the value $h_{\text{crit}}$ where the second quasi-particle appears in the post-quench spectrum.

### 4.3 Large quenches above the threshold: slow oscillations

Above the bound state threshold, i.e. for $h > h_{\text{crit}}$ there are two quasi-particle species of masses (a.k.a. quasi-particle gaps) $m_1$ and $m_2$, and the binding energy $m_2 - 2m_1$ goes to zero when $h$ approaches $h_{\text{crit}}$ from above. As a result, the quench dynamics contains slow oscillations with the frequency $m_2 - 2m_1$, similar to those recently observed in the time evolution of entropies and one-point functions in mass quenches in the $E_8$ field theory [42, 43]. It turns out that in the $E_8$ case entanglement growth is suppressed and iTEBD numerics can be performed for very long times, which allows to extract the frequency with a very high precision. For the $E_8$ theory the source of the long period oscillations is the third quasi-particle, with a mass $m_3 = 1.989\ldots m_1$, which is known exactly in terms of the mass $m_1$ of the lightest quasi-particle due to the integrability of the model [44].

In the longitudinal quenches of the paramagnetic quantum Ising spin chain, entanglement growth is enhanced for $h > h_{\text{crit}}$ by the "dynamical Gibbs effect", so even observing a single period of the oscillation takes quite some effort. Nevertheless, we were able to demonstrate its presence for a large quench from $g = 2.00, h = 0$ to $g = 2.00, h = 0.7$. Although the model is non-integrable, the quasi-particle masses can be computed with sufficient precision from exact diagonalisation [25] resulting in $2m_1 - m_2 = 0.158$ (in units with $J = 1$), leading to a period $T \approx 40$. As shown in Fig. 4.3 this period matches well the oscillations observed in iTEBD evaluation of the magnetisation and entropy time evolutions. Two other examples of slow oscillations present in large longitudinal quenches of the quantum Ising spin chain are shown in Figs. B.7 and B.8.

The same effect can also be observed in the quantum Potts spin chain, as shown in Fig. 4.4 for a quench from $g = 2.00, h = 0$ to $g = 2.00, h = 1.4$, where the quasi-particle masses are again determined from exact diagonalisation following [26] with the result $m_1 = 2.480$ and $m_2 = 4.714$, leading to a period $T \approx 25.62$ which matches the signature in the numerical simulation.

## 5 Conclusions

The existence of a direct relation between relaxation rates and entropy growth after a quantum quench is not just an idea intuitively supported by the quasi-particle picture of the dynamics [1], but a more direct, quantitative relation is suggested by the fact that the growth rate of Rényi entropies can be represented as the relaxation rate for branch-point twist fields related to replica symmetry [22, 43]. Here we examined this relation in the context of quantum quenches on the quantum Ising and Potts spin chains i.e. $q$-state Potts spin chains with $q = 2, 3$.

For transversal quenches (i.e. in the absence of explicit breaking of the symmetry $\mathbb{Z}_q$) we found that the ratio of Rényi entropy growth rates to the magnetisation relaxation rates is universal in the small quench limit

$$\frac{\Gamma_n}{\Gamma} = \frac{1 - 1/q}{1 - 1/n} + \ldots, \quad (n = 2, 3, \ldots), \tag{5.1}$$

with the ellipsis denoting corrections for higher post-quench density. This can be proven explicitly using exact results for the Ising chain, and it is also consistent with the above-mentioned expression of Rényi entropy rates as relaxation rates of branch-point twist fields; we also presented numerical evidence using iTEBD simulations. The value of the ratio can be understood heuristically in terms of the symmetries $\mathbb{Z}_q$ of the magnetisation operator and $\mathbb{Z}_n$ of the branch-point twist fields, however, at this point we lack a fully convincing derivation. One possible way to go is to consider the scaling limit of the transverse field 3-state Potts chain, which is an integrable quantum field theory [45], for which the exact $S$-matrices are known both in

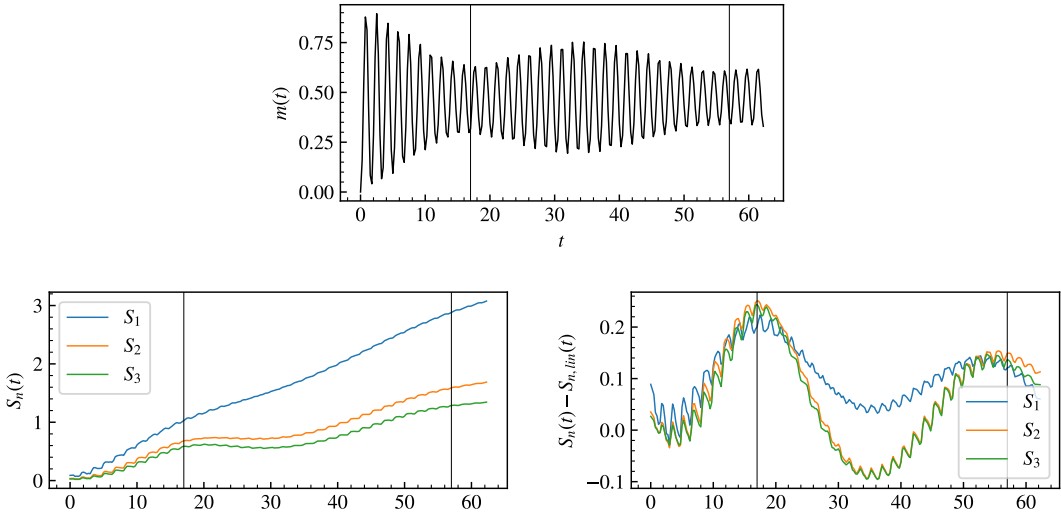

Figure 4.3: Time evolution of various quantities after a large quench $(g_0, h_0) = (2.0, 0.0) \rightarrow (g, h) = (2.0, 0.7)$ in the quantum Ising spin chain. The top left diagram shows the evolution of magnetisation, the top right one the von Neumann entropy ($S_1$) and two of the Rényi entropies ($S_2$ and $S_3$), while the bottom one shows the entropy evolution with the linear trend subtracted. The vertical lines correspond to a single period of the slow oscillation corresponding to the frequency $2m_1 - m_2$. The first line is drawn at the first minimum of the envelope of the magnetisation oscillations while second is drawn at a distance given by the period $T \approx 40$. Note that their position matches quite well the behaviour of the entropy curves as well. Time is measured in units of $1/J$, and the initial entropy was subtracted.

the paramagnetic [46] and ferromagnetic [36] phases. In fact, the parameter $q$ can even be generalised to a real variable between 0 and 4. Therefore it must be possible to repeat the field theory derivation of both the relaxation rate and the Rényi entropy growth rates for the transverse quenches for $q \neq 2$, and directly verify that the universal ratio (3.10) is indeed obtained at the leading order in the post-quench quasi-particle density. However, this requires a rather involved and lengthy calculation (cf. [23]) which is beyond the scope of the present work.

Note that there is no similar universal ratio for the von Neumann entropy rate obtained as the limit $n \rightarrow 1$, which is not surprising as there is no semi-local operator (such as the twist field) to express it as an expectation value.

For quenches involving the longitudinal field the symmetry $\mathbb{Z}_q$ is explicitly broken. In the ferromagnetic phase when the transverse field $g < 1$ this leads to confinement, which can limit the growth of entropy and at the same time leads to persistent oscillations [24], which is consistent with both the entropy rate and the relaxation rate vanishing.

The behaviour in the paramagnetic regime is more interesting. Here we considered the family of quenches starting from the ground state at a transverse field $g > 1$ and switching the longitudinal field $h$ from zero to a finite value. Since the order parameter symmetry $\mathbb{Z}_q$ is explicitly broken, we expect that the universal ratio above is not valid even for small quenches. However, for the Rényi entropy rates we still predict

$$\frac{\Gamma_n}{\Gamma_m} = \frac{1 - 1/m}{1 - 1/n} + \dots, \quad (n, m = 2, 3, \dots), \tag{5.2}$$

which is indeed confirmed by numerical simulations.

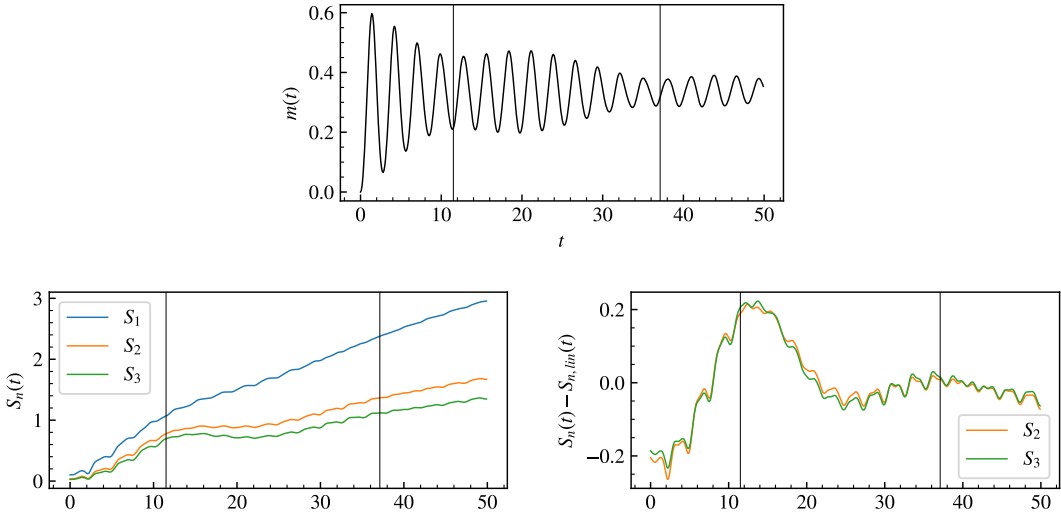

Figure 4.4: Time evolution of various quantities after a large quench $(g_0, h_0) = (2.0, 0.0) \rightarrow (g, h) = (2.0, 1.4)$ in the quantum Potts spin chain. The top left diagram shows the evolution of magnetisation, the top right one the von Neumann entropy ($S_1$) and two of the Rényi entropies ($S_2$ and $S_3$), while the bottom one shows the entropy evolution with the linear trend subtracted. The vertical lines correspond to a single period of the slow oscillation corresponding to the frequency $2m_1 - m_2$. The first line is drawn at the first minimum of the envelope of the magnetisation oscillations while second is drawn at a distance given by the period $T \approx 25.6$. Note that their position matches quite well the behaviour of the entropy curves as well. Time is measured in units of $1/J$, and the initial entropy was subtracted.

Despite the absence of a simple quantitative relation, the growth rates of Rényi entropies and the magnetisation relaxation rate qualitatively follow the behaviour of the growth rate of the von Neumann entropy, as expected from the quasi-particle picture. In particular, all rates show the characteristic non-monotonous behaviour in $h$ due to changes in the quasi-particle spectrum found in [25, 26], i.e. the "dynamical Gibbs effect". This effect is a sudden rise in the entropy growth rate triggered by the appearance of a new quasi-particle in the spectrum, which is attributed to the contribution of species mixing entropy (cf. also [47]). The important implication of our results is that in an experimental implementation of the spin chain the "dynamical Gibbs effect' has a well-defined signature in the behaviour of the magnetisation relaxation rates which is easily observable, in contrast to the entanglement entropy, opening an avenue for the experimental demonstration of the effect.

An interesting observation is that while the ratios of Rényi entropy rates deviate from (5.2) by increasing the quench size parametrised by $h$, the universal ratio is restored to a good precision in a well defined region. This happens approximately at the threshold $h_{\text{crit}}$ for the appearance of the bound state quasi-particles, where the entropy growth is suppressed despite of the finite magnitude of the quench. The theoretical explanation of this finding is an interesting open question.

# Acknowledgements

The authors are grateful to O. Castro-Alvaredo, M. Kormos and J. Viti for discussions and useful comments on the manuscript.

**Funding information**   This work was partially supported by the National Research Development and Innovation Office of Hungary under the postdoctoral grant PD-19 No. 132118 awarded to M.L., and under the research grant K-16 No. 119204 and also within the Quantum Technology National Excellence Program (Project No. 2017-1.2.1-NKP-2017-00001). The work of G.T. was also partially supported by the BME-Nanotechnology FIKP grant of ITM (BME FIKP-NAT).

# A   Details of the iTEBD calculations

The time evolution is computed using the infinite volume time evolving block decimation (iTEBD) algorithm [48]. Using translational invariance, the many-body state is represented as the Matrix Product State (MPS)

$$|\Psi\rangle = \sum_{\ldots,s_j,s_{j+1},\ldots} \cdots \Lambda_o \Gamma_o^{s_j} \Lambda_e \Gamma_e^{s_{j+1}} \cdots |\ldots,s_j,s_{j+1},\ldots\rangle \, ,$$

where $s_j$ spans the local 2-dimensional spin Hilbert space for the quantum Ising spin chain and 3-dimensional spin Hilbert space for the quantum Potts spin chain, $\Gamma_{o/e}^s$ are $\chi \times \chi$ matrices associated with the odd/even lattice site; $\Lambda_{o/e}$ are diagonal $\chi \times \chi$ matrices with the singular values corresponding to the bipartition of the system at the odd/even bond as their entries. The many-body state is initialised to the product state corresponding to pure ferromagnetic states $|\Psi_0\rangle$.

For the quantum Ising spin chain the ground state $|\Psi(0)\rangle$ was construced by time-evolving the initial state $|\Psi_0\rangle$ in imaginary time by the pre-quench Hamiltonian using a second-order Suzuki-Trotter decomposition of the evolution operator with imaginary time Trotter step $\tau = 2.5 \cdot 10^{-4}$, keeping at most $\chi_{max} = 1024$, and $N_{imag} = 300000$ Trotter steps were carried out. The post-quench time evolution was computed for the quantum Ising spin chain by evolving $|\Psi(0)\rangle$ with the post-quench Hamiltonian in real time, again using a second-order Suzuki-Trotter decomposition of the evolution operator with real time Trotter step $\delta t = 2.5 \cdot 10^{-3}$, keeping the same protocol for the singular values. For zooming in the regions around the minima of the entanglement growth we used $\chi_{max} = 512$.

For the quantum Potts spin chain the same procedure was performed with first-order Suzuki-Trotter decomposition with imaginary time Trotter step $\tau = 10^{-3}$ and $\chi_0 = 81$ to obtain the initial state. As the entanglement increases with time, the bond dimension is dynamically updated in order to control the truncation error, with the upper bound $\chi_{\max} = 729$.

# B Further simulations of longitudinal quenches

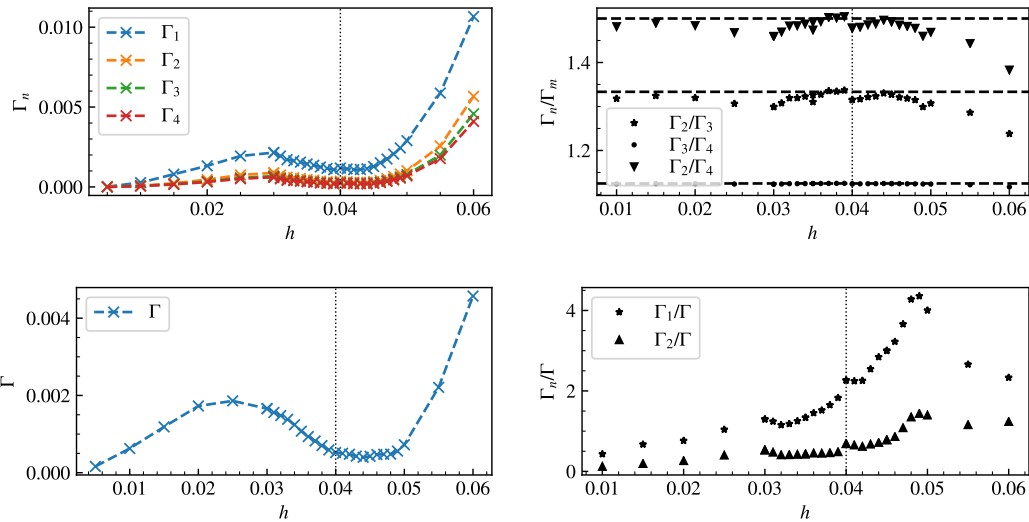

Figure B.1: Left panels: Entropy growth rates $\Gamma_n$ (top) and magnetisation relaxation rate $\Gamma$ (bottom) in units $J = 1$ for longitudinal quenches in the quantum Ising spin chain starting from $g = 1.25$ as a function of the longitudinal coupling. Right: Ratios $\Gamma_n/\Gamma_m$ (top) and $\Gamma_n/\Gamma$ (bottom). The dashed lines represent the universal ratios (4.4) that are restored in the limit $h \to 0$. The vertical line is drawn at the value $h_{\text{crit}}$ where the second quasi-particle appears in the post-quench spectrum.

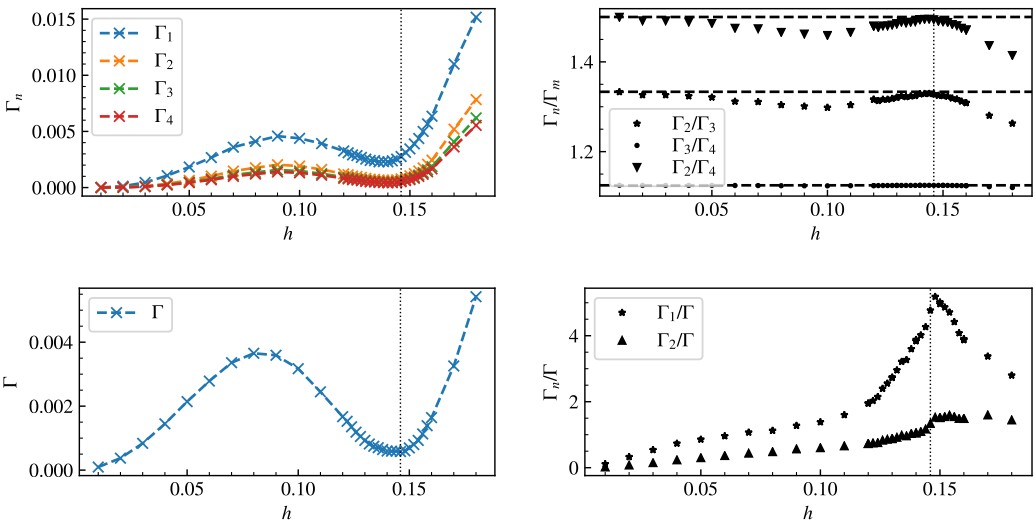

Figure B.2: Left panels: Entropy growth rates $\Gamma_n$ (top) and magnetisation relaxation rate $\Gamma$ (bottom) for longitudinal quenches in the quantum Ising spin chain starting from $g = 1.5$ as a function of the longitudinal coupling, in units with $J = 1$. Right: Ratios $\Gamma_n/\Gamma_m$ (top) and $\Gamma_n/\Gamma$ (bottom). The dashed lines represent the universal ratios (3.10) that are restored in the limit $h \to 0$. The vertical line is drawn at the value $h_{\text{crit}}$ where the second quasi-particle appears in the post-quench spectrum.

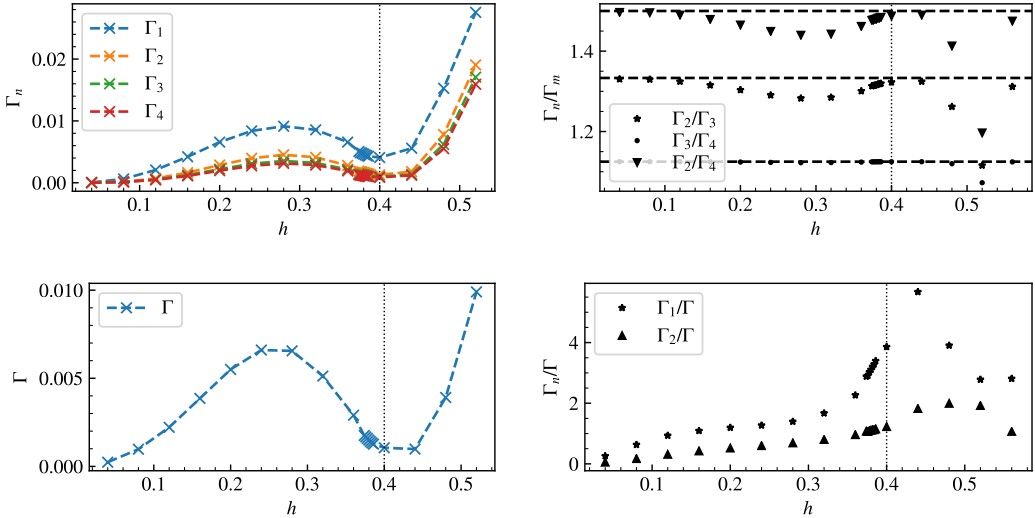

Figure B.3: Left panels: Entropy growth rates $\Gamma_n$ (top) and magnetisation relaxation rate $\Gamma$ (bottom) in units $J = 1$ for longitudinal quenches in the quantum Ising spin chain starting from $g = 2.0$ as a function of the longitudinal coupling. Right: Ratios $\Gamma_n/\Gamma_m$ (top) and $\Gamma_n/\Gamma$ (bottom). The dashed lines represent the universal ratios (4.4) that are restored in the limit $h \to 0$. The vertical line is drawn at the value $h_{\text{crit}}$ where the second quasi-particle appears in the post-quench spectrum.

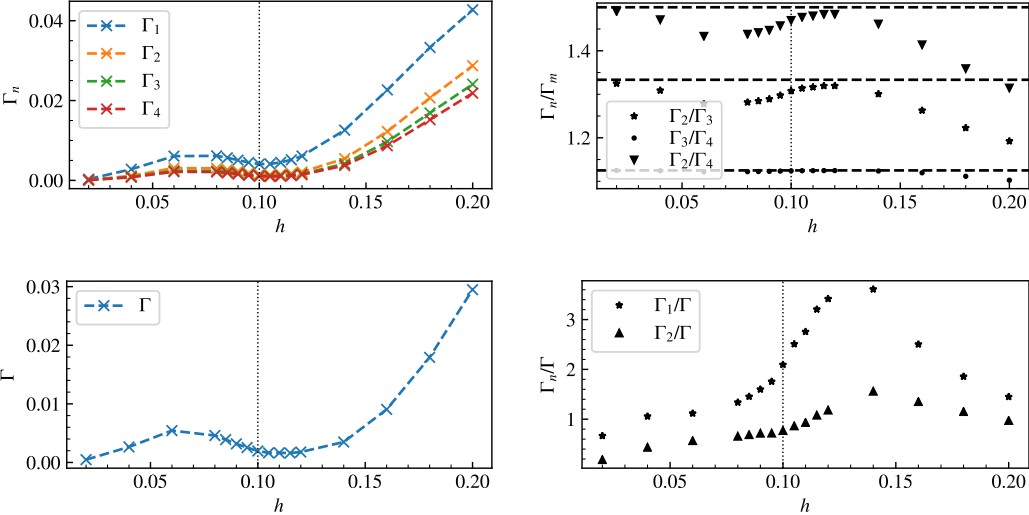

Figure B.4: Left panels: Entropy growth rates $\Gamma_n$ (top) and magnetisation relaxation rate $\Gamma$ (bottom) in units $J = 1$ for longitudinal quenches in the quantum Potts spin chain starting from $g = 1.25$ as a function of the longitudinal coupling. Right: Ratios $\Gamma_n/\Gamma_m$ (top) and $\Gamma_n/\Gamma$ (bottom). The dashed lines represent the universal ratios (4.4) that are restored in the limit $h \to 0$. The vertical line is drawn at the value $h_{\text{crit}}$ where the second quasi-particle appears in the post-quench spectrum.

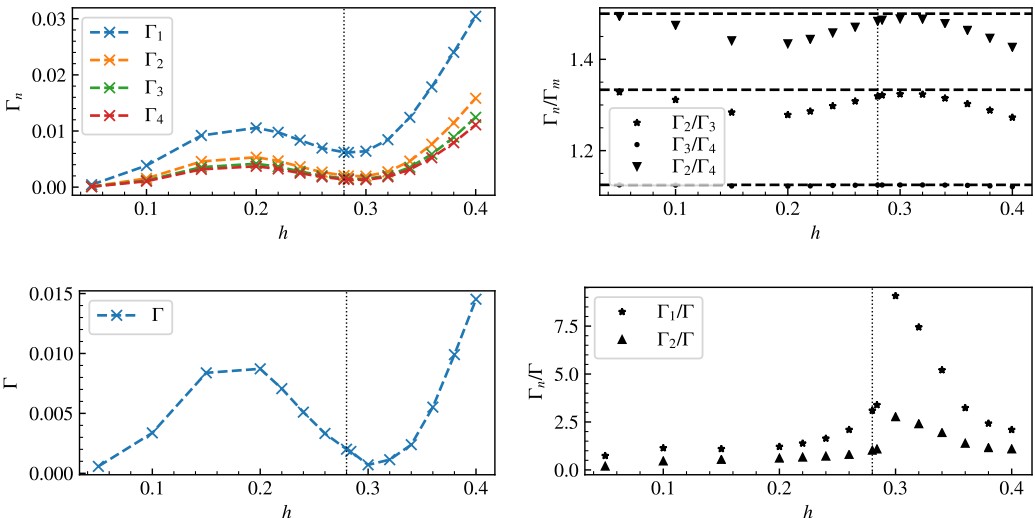

Figure B.5: Left panels: Entropy growth rates $\Gamma_n$ (top) and magnetisation relaxation rate $\Gamma$ (bottom) in units $J = 1$ for longitudinal quenches in the quantum Potts spin chain starting from $g = 1.5$ as a function of the longitudinal coupling. Right: Ratios $\Gamma_n/\Gamma_m$ (top) and $\Gamma_n/\Gamma$ (bottom). The dashed lines represent the universal ratios (4.4) that are restored in the limit $h \to 0$. The vertical line is drawn at the value $h_{\text{crit}}$ where the second quasi-particle appears in the post-quench spectrum.

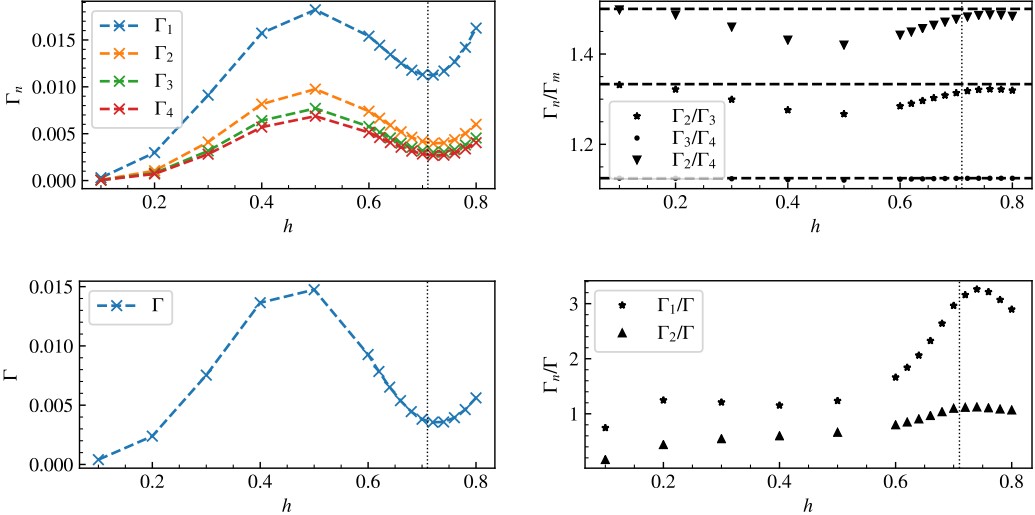

Figure B.6: Left panels: Entropy growth rates $\Gamma_n$ (top) and rmagnetisation relaxation rate $\Gamma$ (bottom) in units $J = 1$ for longitudinal quenches in the quantum Potts spin chain starting from $g = 2.0$ as a function of the longitudinal coupling. Right: Ratios $\Gamma_n/\Gamma_m$ (top) and $\Gamma_n/\Gamma$ (bottom). The dashed lines represent the universal ratios (4.4) that are restored in the limit $h \to 0$. The vertical line is drawn at the value $h_{\text{crit}}$ where the second quasi-particle appears in the post-quench spectrum.

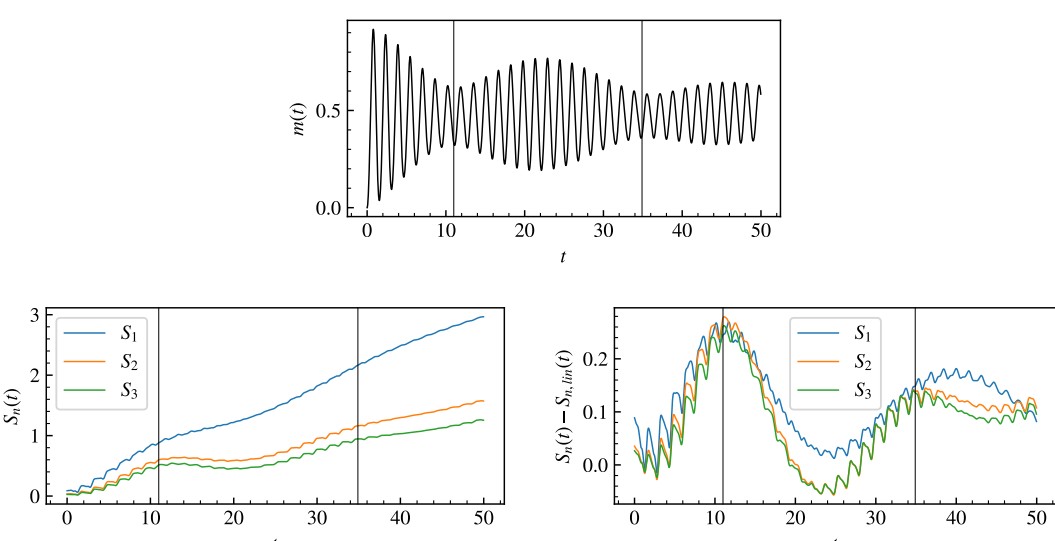

Figure B.7: Time evolution of various quantities after a large quench $(g_0, h_0) = (2.0, 0.0) \to (g, h) = (2.0, 0.8)$ in the quantum Ising spin chain. The top left diagram shows the evolution of magnetisation, the top right one the von Neumann entropy ($S_1$) and two of the Rényi entropies ($S_2$ and $S_3$), while the bottom one shows the entropy evolution with the linear trend subtracted. The vertical lines correspond to a single period of the slow oscillation corresponding to the frequency $2m_1 - m_2$. The first line is drawn at the first minimum of the envelope of the magnetisation oscillations while second is drawn at a distance given by the period $T \approx 23.9$. Note that their position also matches the behaviour of the entropy curves quite well. Time is measured in units of $1/J$, and the initial entropy was subtracted.

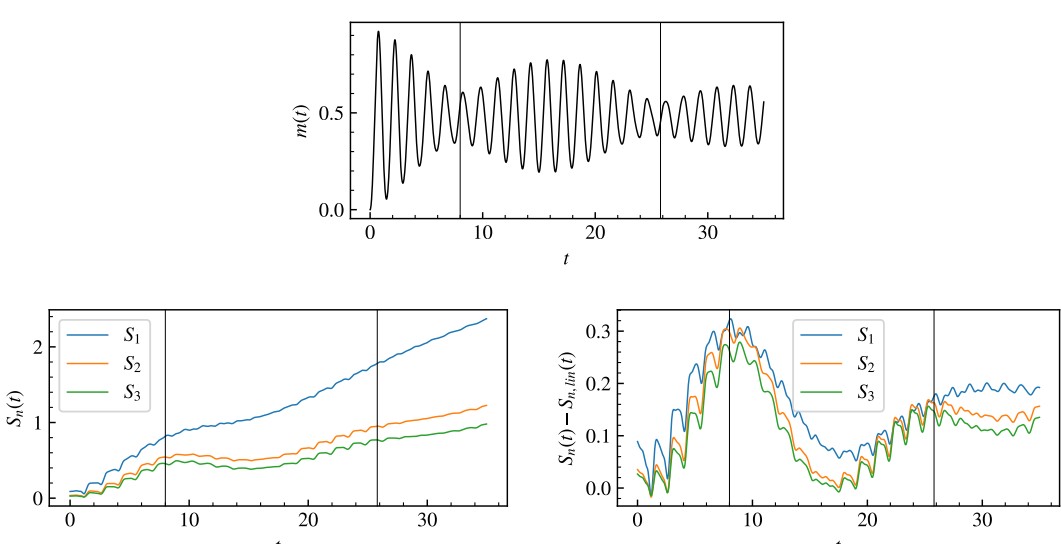

Figure B.8: Time evolution of various quantities after a large quench $(g_0, h_0) = (2.0, 0.0) \rightarrow (g, h) = (2.0, 0.9)$ in the quantum Ising spin chain. The top left diagram shows the evolution of magnetisation, the top right one the von Neumann entropy ($S_1$) and two of the Rényi entropies ($S_2$ and $S_3$), while the bottom one shows the entropy evolution with the linear trend subtracted. The vertical lines correspond to a single period of the slow oscillation corresponding to the frequency $2m_1 - m_2$. The first line is drawn at the first minimum of the envelope of the magnetisation oscillations while second is drawn at a distance given by the period $T \approx 17.8$. Note that their position also matches the behaviour of the entropy curves quite well. Time is measured in units of $1/J$, and the initial entropy was subtracted.

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
