# Peer review of "Relaxation and entropy generation after quenching quantum spin chains"

_SciPost Physics, doi:SciPost Phys. 9, 011 (2020)_

## Round 2 · Referee Report · Olalla Castro-Alvaredo (Referee 1) · 2020-5-11

Strengths

1) The paper addresses a timely and non-trivial problem within the study of the post-quench dynamics of one-dimensional models. 2) Although the paper focuses on the study of particular models, the results obtained suggest a more universal underlying principle still to be fully explored and understood. 3) The connection between entropy growth rate and order parameter decay rate that is found in this work paves the way towards an experimental measurement of entropy growth rate or at least an experimental verification of some key features of this growth rate. 4) The numerical results of the paper provide and explicit observation of the "Dynamical Gibbs Effect".

Weaknesses

1) The scope of the paper is limited to two specific models and although there is a suggestion the results may be more universal that this, there is no full understanding of how the features observed here may generalize to other theories.

Report

In recent years a huge amount of work has been devoted to understanding the dynamical properties of quantum systems (particularly one-dimensional systems) following a quantum quench.

Within this subject two specific problems have attracted much attention: 1) the time-evolution of entanglement measures, such as the von Neumann entropy 2) the time-evolution of expectation values of local fields.

In particular, and especially in the presence of integrability, a frequently observed pattern is linear growth of the von Neumann entropy in conjunction with exponential decay of expectation values of local fields.

In this paper the authors explore this dual behaviour, starting from the premise that the two phenomena are not independent but related to each other when considering the Rényi entropies and the fact that they can be expressed in terms of the expectation value of a branch point twist field. In particular, they provide numerical evidence (and some analytic arguments) for small quenches, that the ratio of the Rényi entropies growth rate and the magnetization decay rate is universal.

In addition they observe that the Rényi entropy growth rate is non-monotonic function of the coupling whose stationary points are very close to those of the von Neumann entropy and whose general features (i.e. the presence of a maximum and a minimum) are closely followed by the decay rate of the magnetization.

Hence experimental measurements of the magnetization could give us information about the monotonicity properties of the entanglement entropy and could provide an experimental signature for the "dynamical Gibbs effect" which is characterized by a minimum of the entropy growth rate, followed by a steep increase associated with the creation of new quasi-particles in the model.

Overall, the paper is well written and contains many interesting analytic and numerical results. To my knowledge it is the first paper that focuses specifically on the quantitative and qualitative relationship between entropy growth rate and magnetization decay rate. In my view it certainly deserves to be published in SciPost.

Requested changes

I have just some very minor comments/suggestions:

1) I noticed that the authors use the notation m(t) and m_0(t) for the magnetization. However they use m and m_0 earlier as masses in the formulae (2.16) and (2.17). I think it is clear from the context that these are two different things, but perhaps since the formulae (2.16) and (2.17) are never really used thereafter, the authors could consider changing their notation for the masses. 2) In the introduction the authors write "The range of validity of this picture includes models with non-interacting quasi-particles, as well as integrable models with fully elastic scattering". Here they are referring to the quasi-particle picture. I think the sentence would be more correct if they add the word "some" before the word "integrable" as not all integrable models with elastic scattering as well described by the quasi-particle picture (see e.g. [38]). 3) Also in the introduction, towards the end, they write "...and the replica symmetry of the Rényi entropy." Although I understand what they mean the sentence does not make much sense. The Rényi entropy itself knows nothing about replica symmetry. The point is rather that replica symmetry is a tool that is used to compute the Rényi entropy. I would replace the sentence by "...and the replica symmetry underlying the computation of the Rényi entropy". 4) After equation (3.9) they write "...the entropy growth rate itself is nothing else than the relaxation rate of the expectation value of the Zn twist field". This is not exactly true (as they explain later in the same page). The correct statement is "...the entropy growth rate itself is nothing else than the relaxation rate of the expectation value of the Zn twist field divided by 1-n". 5) Finally, I find the last paragraph of the conclusion a bit hard to read. I think there are a couple of typos in the first sentence and that sentence is anyway a bit too long. I suggest splitting it in two.

---

## Round 2 · Referee Report · Anonymous (Referee 2) · 2020-6-10

Strengths

1- the claims have been checked numerically 2- the paper in itself is not technical

Weaknesses

1- a rigorous proof of the universality of the phenomenon is missing 2- it is not clear to which class of systems the result applies 3- the paper is not self-contained and it is arguably fully understandable only by experts in the field

Report

This paper investigates the ratio between the rates at which the Rényi entropies grow and the rate at which the longitudinal magnetisation decays in the Ising and in the 3-state Potts spin chains. The authors find that, in the absence of explicit symmetry breaking and in the limit of small quench, the ratios are independent of the quench details. For the Ising model, the authors take advantage of some known analytical results; for the 3-state Potts model, instead, the observation is mainly numerical. In fact, the authors provide also a heuristic argument supporting their conclusion, but a rigorous proof is said to be beyond the scope of the paper. From my point of view, this is a weakness, as a claim of universality requires as many checks as possible.
The second part of the paper is focussed on what the authors call "dynamical Gibbs effect", which is the acceleration of the entanglement entropy growth due to the appearance of a new quasi-particle excitation. The authors find signatures of the effect in the ratios of rates.

In my opinion the publication of these results is premature, but this might be simply related to a different attitude to publication, therefore I will only ask for minor revision. Specifically, despite the paper being non-technical, the explanations rely on concepts, like "semi-locality" and "twist field" that, in my opinion, require an introduction. Thus, I suggest to briefly introduce every concept that is used.

Requested changes

1- Since the terms "semi-local" and "twist field" are used many times, I suggest to (briefly) explain their meaning. 2- Equation (2.9) reports an old conjecture that has been recently shown to be just an excellent approximation. The authors can find the new result in arXiv:2003.09014. This is not relevant for the work, but the authors might want to refer to the correct result.

---

## Round 3 · Referee Report · Olalla Castro-Alvaredo (Referee 1) · 2020-6-26

Strengths

See my 1st Report

Weaknesses

See my 1st Report

Report

When I first reported on this paper I was very positive about it and had only small suggestions for the authors. I can see that they have taken then seriously and have made small improvements.

As before, I think this is a good paper that makes a valuable and timely contribution to the study of the out-of-equilibrium dynamics of entanglement.

Requested changes

None

---

## Round 3 · Author Response

Dear Editor,

hereby I resubmit the revised version of our paper entitled "Relaxation and entropy generation after quenching quantum spin chains". We thank the referees for their suggestions. The changes made in reply are listed below.

Yours faithfully,

Gabor Takacs

---

## Round 3 · List of Changes

To report 1: 1. We changed the notation for the fermion mass to capital M. 2. We agree with the comment, and accordingly added the word "some" in the sentence. 3.-4. Again, we agree with the suggestions for clarification, and changed the sentence accordingly. 5. Indeed, the sentence ended unclear after editing. We split and reformulated it, to make its meaning clear.

To report 2: 1. We added a short subsection 3.3, which defines what we mean exactly by semi-locality, and the parallel between the order parameter and the branch point twist field in this regard. In this process we also added some references. 2. We thank the referee for pointing out the new result. We added the reference and modified the sentence accordingly.

Additionally, we also updated reference [43] (O. A. Castro-Alvaredo et al.: Entanglement Oscillations near a Quantum Critical Point), which has appeared in Phys. Rev. Lett.

---

## Editorial Decision

published